# Role of Intrinsic Subtype Analysis with PAM50 in Hormone Receptors Positive HER2 Negative Metastatic Breast Cancer: A Systematic Review

**DOI:** 10.3390/ijms23137079

**Published:** 2022-06-25

**Authors:** Fabio Canino, Federico Piacentini, Claudia Omarini, Angela Toss, Monica Barbolini, Patrizia Vici, Massimo Dominici, Luca Moscetti

**Affiliations:** 1Division of Medical Oncology, Department of Medical and Surgical Sciences for Children and Adults, University Hospital of Modena, 41125 Modena, Italy; federico.piacentini@unimore.it (F.P.); angela.toss@unimore.it (A.T.); massimo.dominici@unimore.it (M.D.); 2Division of Medical Oncology, Department of Oncology and Hematology, University Hospital of Modena, 41125 Modena, Italy; claudia.omarini@gmail.com (C.O.); barbolini.monica@gmail.com (M.B.); moscetti.luca@aou.mo.it (L.M.); 3Gruppo Oncologico Italiano per la Ricerca Clinica (GOIRC), 43100 Parma, Italy; 4Department of Phase IV Clinical Trials, IRCCS Regina Elena National Cancer Institute, 00144 Rome, Italy; patrizia.vici@ifo.it

**Keywords:** intrinsic subtype, PAM50 assay, metastatic breast cancer, HR+/HER2-, endocrine therapy

## Abstract

Endocrine therapy (ET), associated with CDK 4/6 inhibitors, represents the first choice of treatment for HR+/HER2- metastatic breast cancer (mBC). Primary or secondary endocrine resistance could develop; however validated biomarkers capable of predicting such a conditions are not available. Several studies have shown that HR+/HER2- mBC comprises five intrinsic subtypes. The purpose of this systematic review was to analyze the potential correlations between intrinsic subtype, efficacy of treatment, and patient outcome. Five papers that analyzed the intrinsic subtype with PAM50 assay in patients (pts) with HR+/HER2- mBC treated with ET (alone or in combination) within seven phase III clinical trials (EGF30008, BOLERO-2, PALOMA-2,3, MONALEESA-2,3,7) were identified. Non-luminal subtypes are more frequent in endocrine-resistant pts and in metastatic sites (vs. primary tumors), have less benefit from ET, and worse prognosis. Among these, HER2-enriched subtypes are similar to HER2+ tumors and benefit from the addition of anti-HER2 agents (lapatinib) and, for less clear reasons, of ribociclib (unconfirmed data for palbociclib and everolimus). Basal-like subtypes are similar to triple-negative tumors, making them more sensitive to chemotherapy. The intrinsic subtype is also not static but can vary over time with the evolution of the disease. Currently, the intrinsic subtype does not play a decisive role in the choice of treatment in clinical practice, but has potential prognostic and predictive value that should be further investigated.

## 1. Introduction

Hormone receptor-positive (HR+) metastatic breast cancer (mBC) represents a wide and heterogeneous group of cancers, with different clinical characteristics and prognosis. Currently, excluding the expression of the estrogen (ER) and progesterone (PgR) receptors and the HER2 protein, no other validated biomarkers exist that can guide the choice of treatment in this setting, which is still based on patient and disease characteristics.

About 6% of mBCs are de novo diseases, usually sensitive to upfront endocrine/combined therapies. In the other cases they are systemic relapses of a disease treated in the early setting and already exposed to endocrine therapy (ET) [1]. Depending on the time of onset of the recurrence, primary or secondary resistance to ET can be identified [2], however the biological reasons are still far from being clearly identified.

To better understand this feature, in the last two decades many studies have focused on the microarray evaluation of gene expression patterns in mBCs, identifying five intrinsic subtypes: luminal A (LumA), luminal B (LumB), HER2-enriched (HER2E), basal-like and normal-like, with significant differences in terms of risk factors, incidence, prognosis, and responses to systemic therapies [3].

The prediction analysis of microarray (PAM50) is one of the currently available assays capable of identifying the intrinsic subtypes of BC through the quantitative measurement of the expression of 50 different genes [4].

All five subtypes are represented within HR+/HER2-BC and, although they share similar clinical–pathological characteristics and receive the same treatment, each one has different peculiarities. Compared to the LumA and -B subtypes, the non-luminal subtypes are characterized by lower differentiation, higher expression of proliferation markers (i.e., Ki67), and lower levels of genes regulated by hormonal pathways. Specifically, the HER2E subtypes are similar to HER2+/HER2E tumors but lack the amplification/ overexpression of the HER2 amplicon (ERBB2 and GRB7); on the other hand, the basal-like subtypes are similar to triple-negative breast cancers [5,6]. Studies in the adjuvant setting have shown that HR+/HER2- BC with non-luminal subtype are associated with endocrine resistance [7,8], chemo-sensitivity [9], and poorer prognosis [10].

As the identification of intrinsic subtypes within HR+/HER2- BC has potential clinical implications, studies have also been conducted in the advanced setting.

This study is a systematic review of the literature. Its aim was to collect data on the identification of intrinsic subtypes by the PAM50 assay in pts with HR+/HER2- mBC who have received first-line or subsequent ET alone or as a combination treatment in phase III clinical trials in order to evaluate possible correlations between intrinsic subtype, treatment efficacy, and patient outcome.

## 2. Methods

### 2.1. Data Sources and Search Strategy

A systematic literature search of the electronic databases Medline and Web of Science was conducted. The search identified studies were published as full-text articles and analyzed the intrinsic subtype with the PAM50 assay on histological samples from pts with HR+/HER2- mBC, both endocrine-sensitive and endocrine-resistant, treated with ET alone or in combination in randomized and controlled phase III clinical trials. The results of these clinical trials were themselves already known and published as full-text articles.

No restrictions in terms of language or year of publication were applied. The final date for the database search was 31 December 2021.The search strategy was developed using the patient, intervention, comparator, and outcome (PICO) framework. The terms used for the search strategy were as follow: “metastatic breast cancer”, “hormone receptors positive”, “intrinsic subtypes”, and “PAM50”. Boolean operators were used to connect specific search keywords for each database and other free-text terms. The specific rules and vocabulary of the database were used. The search strategy was designed by two authors (F.C. and L.M.) and discussed with the other authors. The references listed in all the identified papers were assessed to find any additional eligible studies. The present analysis was conducted according to the Preferred Reporting Items for Systematic Reviews and Meta-Analyses (PRISMA) [11].

### 2.2. Elegibility Criteria

The following types of studies were included in this systematic review: (1) studies that performed intrinsic subtype analysis with the PAM50 array on histological samples belonging to pts with HR+/HER2- mBC, both hormone sensitive and hormone resistant, treated with ET alone or in combination in a randomized and controlled phase III clinical trial, and (2) randomized and controlled phase III clinical trials comparing therapeutic regimens that included ET in both experimental and control arms, concluded and published as full-text articles.

The following types of studies were excluded from this systematic review: (1) studies performing intrinsic subtype analysis with the PAM50 array on histological samples belonging to pts with different characteristics, including: early breast cancer, HR- and/or HER2+ or triple negative mBC; (2) studies performing intrinsic subtype analysis with the PAM50 array on histological samples belonging to pts with HR+/HER2- mBC, treated outside of completed, randomized, controlled phase III clinical trials, such as in: ongoing randomized and controlled phase III clinical trials, phase I/II clinical trials, or non-randomized clinical trials; or (3) concluded, randomized, and controlled phase III clinical trials comparing therapeutic regimens that did not include ET in both experimental and control arms.

Two investigators (F.C. and L.M.) independently extracted the data from all the included studies. The following variables were collected: first author, name of the trial, year of publication, overall sample size, description of standard and experimental treatments, study results and impact on clinical practice, number of histological samples analyzed by intrinsic subtype, distribution of intrinsic subtypes for each study, correlation analysis of the efficacy of the experimental treatment and/or patient outcome compared to intrinsic subtype (if available), and any other observations of the authors related to the intrinsic subtype.

## 3. Results

With the terms used for the search strategy, 119 publications were identified for initial eligibility screening. Based on the information found in their titles and/or abstracts, 26 studies were excluded because they were duplicates and 42 were deemed not relevant to the review. Of the remaining 51 publications, 46 were excluded because they did not meet all the inclusion criteria: 11 were reviews, 5 were abstracts or posters, 11 were non-interventional clinical trials, 16 were clinical trials involving populations other than those being analyzed in this review (early breast cancer, HR- and/or HER2 positive, or triple-negative breast cancer), 1 was a phase II study, and 2 studies did not include ET in the therapeutic regimens investigated. Finally, a total of 5 studies were identified and deemed suitable for our systematic review (Figure 1).

In the five studies considered, the intrinsic subtypes of 2999 histological samples were analyzed. These samples belonged to a total of 5263 pts with mBC treated in seven different randomized controlled phase III clinical trials: EGF30008, BOLERO-2, PALOMA-2,3, and MONALEESA-2,3,7. Below we report the results of each individual study.

In 2016, Prat et al. [12] published the first work of this kind, using the PAM50 assay to analyze the intrinsic subtype of tumor samples of pts treated in the **EGF30008** [13]. It was a phase III trial that evaluated the efficacy of letrozole +/− lapatinib in 1286 postmenopausal pts with HR+ advanced breast cancer (ABC), with both HER2-positive and HER2-negative disease, stage III or IV, previously untreated in the metastatic setting. Prior neoadjuvant (nadj)/adjuvant (adj) ET was allowed, and extensive symptomatic visceral disease was excluded. The study showed that the addition of lapatinib to letrozole resulted in a reduction in the risk of progression in the HR+/HER2+ pts (HR 0.71; 95% CI 0.53–0.96; *p* = 0.019) with a median progression-free survival (mPFS) of 8.2 months (vs. 3 in the control arm), but no benefit was found in the HR+/HER2- population. Eight hundred twenty-one samples of 1286 pts were included in the final analysis (85.7% of the samples were from primary tumor, 14.7% from metastatic tissue). Six hundred forty-four were HR+/HER2- (distribution of intrinsic subtypes: LumA 52%, LumB 30%, HER2E 3%, basal-like 3%, normal-like 12%) and 157 HR+/HER2+ (distribution of intrinsic subtypes: LumA 27%, LumB 29%, HER2E 29%, basal-like 4%, normal-like 12%). Compared to the HR+/HER2+ group, the HR+/HER2- group had a higher proportion of LumA (52% vs. 27%) and a smaller proportion of HER2E (3% vs. 29%), whereas the percentages of LumB, basal-like and normal-like were similar. There were no differences in the distribution of intrinsic subtypes based on the number of metastases, type of metastases (visceral/bone only), or treatment arm. However, a higher proportion of non-luminal subtypes was observed in pts who relapsed during adj ET or within six months after treatment discontinuation, compared to those who relapsed more than six months after discontinuation or had never undergone adj ET (17% vs. 9% of non-luminal subtypes, of which 70% were HER2E). In the HER2- group, LumA recorded the best outcomes in terms of progression-free survival (PFS) (16.9 vs. 11 vs. 4.7 vs. 4.1 months) and overall survival (OS) (45 vs. 37 vs. 16 vs. 23 months) compared to the other subtypes, regardless of clinical–pathological variables with recognized prognostic significance. In addition, HR+/HER2- pts with the HER2E subtype benefited from the addition of Lapatinib to therapy (mPFS 6.49 vs. 2.6 months, HR 0.238; 95% CI 0.066–0.863, *p* = 0.03).

A similar study was published by Prat et al. in 2019 [14] and analyzed the intrinsic subtype with PAM50 assay of pts treated in **BOLERO-2** [15], a phase III trial evaluating the efficacy of exemestane +/− everolimus in 724 postmenopausal pts with HR +/HER2- ABC previously treated with non-steroidal aromatase inhibitors (NSAI) in the metastatic setting. The study showed that the addition of everolimus to exemestane increased PFS in the experimental group (mPFS 7.8 vs. 3.2 months; HR 0.45; 95% CI 0.38–0.54; *p* < 0.0001) [16], leading to the approval of the drug combination in this setting. Two hundred sixty-one samples from 724 pts were included in the final analysis of the study by Prat et al. (80.7% from primary tumor, 19.3% from metastasis). The intrinsic subtypes were distributed as follows: LumA 46.7%, LumB 15.7%, HER2E 21.5%, basal-like 1.9%, normal-like 14.2%. luminal subtypes (A and B) had a better mPFS than non-luminal subtypes (mPFS 6.7 vs. 5.2 months; adjusted HR 0.66; 95% CI, 0.47–0.94; *p* = 0.020). The HER2E subtype had worse prognosis, with lower PFS than non-HER2E (5.2 vs. 6.2 months; adjusted HR 1.53; 95% CI 1.07–2.19; *p* = 0.019), and was present more frequently in metastatic tissue than in primary tumor (32.0% vs. 18.7%; *p* = 0.038). No specific analyses were performed on the basal-like subtype, as the number was limited (5 pts). The addition of everolimus to exemestane resulted in a PFS improvement in all subtypes: In HER2E this difference was less evident (5.8 vs. 4.1 months; adjusted HR 0.49; 95% CI, 0.26–0.90; *p* = 0.034) compared to non-HER2E subtypes (8.7 vs. 4.1 months; adjusted HR 0.37; 95% CI, 0.26–0.51; *p* < 0.0001). The interaction between the HER2E subtype and the benefit of everolimus over PFS was not statistically significant (*p* = 0.433). This result was also valid for the whole subgroup of non-luminals (*p* = 0.534).

Subsequent studies evaluated the prognostic/predictive value of intrinsic subtypes in HR+/HER2- ABC treated with ET + cyclin dependent kinase 4/6 inhibitors (CDK4/6i).

Turner et al. in 2019 [17] retrospectively analyzed the intrinsic subtype with the PAM50 assay in pts treated in **PALOMA-3** [18], a phase III trial evaluating the efficacy of adding palbociclib to fulvestrant in 521 endocrine-resistant pts with HR+/HER2- ABC. The study showed that the addition of palbociclib to fulvestrant increased PFS in the experimental group (mPFS 9.5 vs. 4.6 months; HR 0.46; 95% CI 0.36–0.59; *p* < 0.0001, after a median follow up of 8.9 months) [19], leading to the approval of the drug combination in this setting. Three hundred two samples from 521 pts were included in the final analysis with PAM50 assay (53% from primary tumor; 47% from metastasis). The distribution of the subtypes was as follows: LumA 44%, LumB 30.8%, HER2E 20.9%, basal-like 1.7%, normal-like 2.6%. The LumA subtype had better PFS than the other subtypes. The addition of palbociclib improved PFS in all subgroups, with a greater impact on the luminal subtypes (mPFS in LumA: 16.6 vs. 4.8 months; HR 0.41; 95% CI 0.25–0.66; mPFS in LumB: 9.2 vs. 3.5 months; HR 0.64; 95% CI 0.38–1.09) compared to non-luminals, considered as a single subgroup in this study (mPFS in non-luminal: 9.5 vs. 5.5 months; HR 0.58; 95% CI 0.34–0.99; *p* = 0.0451). The trial did not show any association between the intrinsic subtype and palbociclib efficacy. However, the study showed that high levels of cyclin E (CCNE1) mRNA correlated with greater resistance to treatment with palbociclib. In an exploratory analysis it appeared that CCNE1 mRNA levels are higher in the non-luminal and LumB subtypes than in LumA.

Finn et al. [20] analyzed the intrinsic subtype in pts treated in **PALOMA-2** [21], a phase III trial that evaluated the efficacy of adding palbociclib to letrozole in 666 endocrine-sensitive pts with HR+/HER2- ABC. Adding palbociclib to letrozole increased PFS in the experimental group (27.6 vs. 14.5 months; HR 0.56; 95% CI 0.46–0.69; *p* < 0.001, after a median follow up of 38 months) [22], leading to approval of the use of the combination in clinical practice. In the final analysis 455 samples were included out of 666 pts treated. The distribution of intrinsic subtypes (using the absolute intrinsic molecular subtyping (AIMS) algorithm) was as follows: LumA 50.3%; LumB 29.7%; HER2E 18.7%; basal-like 0.5%; normal-like 0.9%. The addition of palbociclib increased survival in all subtypes, impacting the luminal subtypes more, even if the observation was heavily limited by the small sample.

In one of the most recent studies, Prat et al. [23] evaluated the association between intrinsic subtypes, prognosis, and treatment benefit in PFS and overall response rate (ORR) in pts treated in the MONALEESA trials.

**MONALEESA 2** [24] was a phase III study that evaluated the addition of ribociclib to letrozole in 668 post-menopausal endocrine-sensitive pts with HR+/HER2- ABC. The addition of ribociclib to letrozole increased the PFS in the experimental group (25.3 vs. 16 months; HR 0.57; 95% CI 0.46–0.70; *p* < 0.001, after a median follow up of 26.4 months) [25].

**MONALEESA 3** [26] was a phase III study that evaluated the addition of ribociclib to fulvestrant in 726 post-menopausal endocrine-resistant pts with HR+/HER2- ABC. The addition of ribociclib to fulvestrant resulted in an increase in PFS (20.5.3 vs. 12.8 months; HR 0.59; 95% CI 0.48–0.73; *p* < 0.001, after a median follow up of 17 months) and OS (57.8% vs. 45.9% after 42 months of follow up; HR 0.72; 95% CI 0.57–0.92; *p* = 0.0045) [27].

**MONALEESA 7** [28] was a phase III study that evaluated the addition of ribociclib to ET (tamoxifen or aromatase inhibitors + LHRH analogue) in 672 pre- or peri-menopausal endocrine-sensitive patients with HR+/HER2- ABC. The addition of ribociclib to ET resulted in an increased PFS (23.8 vs. 13 months; HR 0.55; 95% CI 0.44–0.69; *p* < 0.001, after a median follow up of 26.4 months) and OS (70.2% vs. 46% after 42 months of follow up; HR 0.71; 95% CI 0.54–0.95; *p* = 0.0097) [29].

These studies led to the approval of the use of ribociclib in clinical practice in association with hormone therapy in both endocrine-sensitive and endocrine-resistant pts, regardless of menopausal state. One thousand one hundred sixty tumor tissue samples (72% from primary tumor; 28% from metastasis) of 2066 pts treated in the MONALEESA trials were collected. The distribution of subtypes analyzed with PAM50 assay was as follows: LumA 46.7%, LumB 24.0%, HER2E 12.7%, basal-like 2.6%, normal-like 14.0%. Compared to primary tumor samples, the HER2E (16.8% vs. 11.2%) and LumB (30.5% vs. 21.7%) subtypes were more frequent in metastases, whereas the LumA subtypes (38.1% vs. 49.9%) were less frequent. LumA had better PFS than other subtypes: LumB, normal-like, HER2E, and basal-like showed 1.4, 2.3, 1.3, and 4.0 times higher risks of disease progression than LumA, respectively. In both treatment arms, median PFS differed across intrinsic subtypes. The addition of ribociclib resulted in an improvement in PFS in all subtypes, except basal-like. This benefit was greater in HER2E, which had a worse prognosis with ET alone. Specifically, comparing the PFS in the experimental vs. control arm for each intrinsic subtype resulted in the following: LumA 29.6 vs. 19.48 months (HR 0.63; 95% CI 0.49–0.83; *p* = 0.0007), LumB 22.21 vs. 12.85 months (HR 0.52; 95% CI 0.38–0.72; *p* < 0.0001), HER2E 16.39 vs. 5.52 months (HR 0.39; 95% CI 0.25–0.60; *p* < 0.0001), basal-like 3.71 vs. 3.58 months (HR 1.15; 95% CI 0.46–2.83; *p* = 0.77), normal-like 22.34 vs. 11.10 months (HR 0.47; 95% CI 0.30–0.72; *p* < 0.001). The addition of ribociclib resulted in a marked improvement in overall response rate (ORR), especially in the LumB (51.9% vs. 29.8%; *p* < 0.001) and HER2E subtypes (40% vs. 9.6%; *p* < 0.001).

## 4. Discussion

The analyzed studies confirm that the intrinsic subtype in HR+/HER2- mBC represents a potential prognostic and predictive biomarker of response to some types of therapy. The distribution and outcome of each intrinsic subtype in the analyzed trials are summarized in Table 1.

The luminal subtypes are the most frequent (62–82%) and have the best PFS and OS, independently of the other clinical–pathological variables with recognized prognostic significance.

Non-luminal subtypes appear to benefit less from ET than luminal subtypes. In the analysis of EGF30008 [12], the percentage of non-luminal was higher in pts relapsed during or within six months after discontinuation of adj ET (17%) compared to pts who relapsed more than six months after discontinuation or who never received adj ET (9%). In the analysis of BOLERO-2 [14], PALOMA-2 [17], and PALOMA-3 [20], non-luminal subtypes had less advantage in PFS from the addition of everolimus or palbociclib to ET, respectively. In the latter case, it was shown that non-luminal subtypes expressed higher levels of cyclin E (CCNE1) mRNA, known to be a biomarker of resistance to palbociclib. However, the authors pointed out that the analysis was only exploratory and other studies would be necessary to acquire more consistent data [17]. In a recent analysis of the YOUNG PEARL trial [30], which compared the efficacy of palbociclib + exemestane + LHRH analogue vs. capecitabine in pre-menopausal women with HR+/HER2- mBC not previously treated with aromatase inhibitors, Lee et al. confirmed better PFS in luminal vs. non-luminal subtypes in pts receiving combined treatment, excluding those with a pathogenetic BRCA2 mutation, characterized by worse prognosis regardless of subtype. However, this finding was not confirmed by the analysis of pts treated in MONALEESA trials [23], in which the addition of ribociclib resulted in an increase in PFS in all subtypes excluding basal-like, and furthermore, HER2E appeared to be the subtype that benefited most from the addition of ribociclib, although it is known to have a worse prognosis and show greater endocrine resistance than the luminal subtypes [7,8] (differences in PFS between experimental and control groups divided by intrinsic subtype were as follows: HER2E 10.8 months, LumA 10.1 months, LumB 9.4 months, basal-like 0.2 months, normal-like 11.24 months). The advantage of ribociclib in this subtype could be related to the combination of cell-cycle inhibition, resensitization to endocrine therapy, and immunomodulation [23]. More information on this could come from the ongoing HARMONIA trial (ClinicalTrials.gov Identifier: NCT05207709), the first prospective Phase III trial to enroll pts selected by RNA-based molecular subtyping of their tumors and the first to directly compare two CDK4/6 inhibitors (ribociclib and palbociclib) and chemotherapy (paclitaxel) in pts with HR+/HER2- and HER2E or basal-like ABC. The primary objective of this study is to demonstrate that the combination of ribociclib + ET (letrozole or fulvestrant) is superior to palbociclib + ET (letrozole or fulvestrant) in prolonging PFS in pts with HR+/HER2- and HER2-E ABC. In addition, the trial includes an exploratory cohort of pts with HR+/HER2- and basal-like ABC treated with paclitaxel, given the suggested lack of efficacy of the combinations of CDK4/6i and ET in this subgroup.

Another relevant topic is the variability of the intrinsic subtype over time in the course of the disease. The possible discrepancy in immunophenotype between primary tumor and metastatic tissue, and its impact on treatment, has long been known. Guarneri et al., compared tissue samples from primary tumor vs. metastasis in 75 pts with ABC, showed a change in HER2 and HR expression from positive to negative in 2.7% and 12% of cases, respectively, and from negative to positive in 13.3% and 9%, respectively; for the latter this resulted in a new line of treatment with HER2 inhibitors or ET that was previously not possible. The authors concluded that performing a re-biopsy of metastatic sites, when feasible, is advisable to increase available treatment options [31]. Given that the immunophenotype function as a surrogate for the BC molecular classification [32], the intrinsic subtype may also change as the disease evolves. In fact, another important finding emerging from these studies is that the HER2E subtype was more frequent in advanced disease (3–20%) than in early disease (3–10%) [3,5,6], but also in metastatic tissue compared to primary tumor, as shown in the analysis of the pts of the BOLERO-2 (32.0% vs. 18.7%) [14] and MONALEESA trials (16.8% vs. 11.2%) [23]. This could be related both to the selection criteria of the pts in the various trials, but also to the evolution of the tumor clones following the treatments for advanced disease. Cejalvo et al. [33] compared the analysis of intrinsic subtypes, according to PAM50 assay, from primary tumor vs. metastasis in 123 tissue samples from pts with ABC: the study demonstrated high concordance between the two tissue types for LumB, HER2E, and basal-like; on the contrary, for LumA it was observed that 40.4% became LumB and 14.9% became HER2E. In addition, about 10–15% of LumA and LumB at diagnosis changed to the HER2E subtype after the first relapse. This finding agreed with studies previously conducted in the advanced setting, in which the rate of HER2E was higher than that recorded in studies in the adjuvant setting, especially considering that the pts in BOLERO-2 [15] and MONALEESA-3 [26] were endocrine resistant. The transition from a luminal subtype to a HER2E can therefore be an expression of acquired endocrine resistance in a tumor that was previously endocrine sensitive. This subtype variation over time could explain why LumA had better prognosis than all other subtypes in the analysis of the EGF30008 trial [12], whereas there were no substantial differences in prognosis between LumA and B in the analysis of BOLERO-2 [14]. In the first case, the pts had not previously been treated for metastatic disease, whereas in the second case pts were endocrine resistant by definition and PAM50 assay of about 80% of the samples was performed on the primary tumor tissue. It is possible that some of the BOLERO-2 pts who originally had a LumA turned into LumB or another subtype after relapse; therefore it is likely that if metastatic tissue had been available for all pts the distribution of intrinsic subtypes would have been different.

Finally, the analysis of these studies highlights the different behavior of non-luminal subtypes. From a biological point of view, HR+/HER2- and HER2E tumors are much more similar to HER2 + HER2E tumors, even if the former does not have amplification/overexpression of the HER2 amplicon (ERBB2 and GRB7) [5]. The driver of these subtypes could be the epidermal growth factor receptor (EGFR), which would explain the PFS advantage of adding lapatinib to letrozole in the HR+/HER2- and HER2E population recorded in the analysis of pts treated in EGF30008 [12]. In this regard, the SOLTI-1718 NEREA trial (ClinicalTrials.gov identifier: NCT04460430) is evaluating the efficacy of adding neratinib to ET (fulvestrant, exemestane, tamoxifen) in HR+/HER2- and HER2E ABC with or without previous treatment with CDK4/6, PI3K, mTOR, or AKT inhibitors. The main objective of this trial is to demonstrate the efficacy of adding neratinib to ET in improving PFS in this subgroup. Moreover, HR+/HER2- and basal-like tumors are biologically more similar to triple-negative breast cancers, characterized by increased expression of cyclin E1, EGFR, and p16 and low expression of ER1, PGR1, and Forkhead box1. In the analysis of the MONALEESA trials [23], the basal-like subgroup was the only one that did not have a benefit in PFS from the addition of ribociclib to ET. Given the likely low sensitivity to ET and the potential sensitivity to chemotherapy, as previously mentioned, this subgroup will undergo chemotherapy with paclitaxel in the exploratory cohort of the HARMONIA trial.

The claudin-low subtype was not specifically analyzed in any of the articles discussed. Claudin-low is a more recently identified intrinsic subtype of BC characterized by low expression of cell–cell adhesion molecules (claudins 3, 4, and 7; occluding; and E-cadherin) highly enriched in mesenchymal tracts and stem-cell characteristics. It shares many characteristics with the basal-like subtype, and together they account for the majority of TNBC [34]. Future studies will be needed to deepen our knowledge of this subgroup.

## 5. Conclusions

Currently, the available data are not robust enough to suggest the routine use of intrinsic subtype analysis in HR+/HER2- mBC treatment choice.

However, the studies carried out in recent years and discussed in this systematic review show the potential prognostic and predictive value of intrinsic subtype determination. Non-luminal subtypes showed worse prognosis than luminals, but may benefit from treatments other than the current standard of care, represented by the CDK4/6i + ET combination. 

Prospective randomized trials are underway (SOLTI-1718 NEREA and HARMONIA) to explore potential new therapeutic approaches.

## Figures and Tables

**Figure 1 ijms-23-07079-f001:**
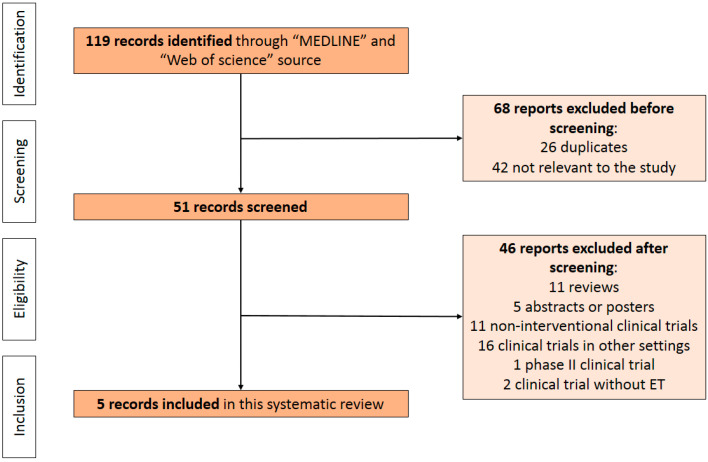
PRISMA flowchart summarizing the process to identify eligible studies.

**Table 1 ijms-23-07079-t001:** Distribution and outcome by intrinsic subtype in pts from (**A**) EGF30008 trial; (**B**) PALOMA-2 trial; (**C**) BOLERO-2 trial; (**D**) PALOMA-3 trial; (**E**,**F**) MONALEESA trial. EXE: exemestane; EVE: everolimus; FULV: fulvestrant; LETRO: letrozole; PALB: palbociclib; RIBO: ribociclib; ET: endocrine therapy; PBO: placebo; NA: not available.

**A**							
	**EGF30008 [13]** **HR+/HER2- pts (n = 644)**						
	**% (n°)** **Overall Population**		**mPFS (Months)** **Overall Population**	**mPFS (Months)** **ET + LAPATINIB**		**mPFS (Months)** **ET + PBO**	
**Lum A**	52 (335)		16.9 (95% CI, 14.1–19.9)	NA		NA	
**Lum B**	30 (196)		11.0 (95% CI, 9.6–13.6)	NA		NA	
**HER2E**	3 (16)		4.7 (95% CI, 2.7–10.8)	NA		NA	
**Basal-like**	3 (21)		4.1 (95% CI, 2.5–13.8)	NA		NA	
**Normal-like**	12 (76)		NA	NA		NA	
**B**							
	**PALOMA-2 [21]** **All pts (n = 455)**						
	**% (n°)** **Overall Population**		**mPFS (Months)** **Overall Population**	**mPFS (Months)** **LETRO + PALB**		**mPFS (Months)** **LETRO + PBO**	**HR**
**Lum A**	50.3 (229)		NA	30.4		17.0	0.55 (95% CI, 0.39–0.77; *p* = 0.0005)
**Lum B**	29.7 (135)		NA	16.9		11.0	0.51 (95% CI, 0.34–0.77; *p* = 0.0019)
**HER2E**	18.7 (85)		NA	NA		NA	NA
**Basal-like**	0.5 (2)		NA	NA		NA	NA
**Normal-like**	0.9 (4)		NA	NA		NA	NA
**C**							
	**BOLERO-2 [15]** **All pts (n = 261)**						
	**% (n°)** **Overall Population**	**% (n°)** **Primary Tumors** **(n = 209#)**	**% (n°)** **Metastasis** **(N = 50#)**	**mPFS (Months) Overall Population**	**mPFS (Months)** **EXE + EVE**	**mPFS (Months)** **EXE + PBO**	**HR**
**Lum A**	46.7 (122)	48.3 (101)	40.0 (20)	6.2 (95% CI, 4.37–8.31)	8.3 (95% CI, 5.59–11.10)	4.1 (95% CI, 2.63–5.26)	0.39 (95% CI, 0.25–0.61; *p* < 0.0001)
**Lum B**	15.7 (41)	14.8 (31)	20.0 (10)	5.4 (95% CI, 4.04–8.05)	6.8 (95% CI, 4.11–11.70)	2.8 (95% CI, 1.48–7.13)	0.69 (95% CI, 0.34–1.39; *p* = 0.349)
**HER2E**	21.5 (56)	18.7 (39)	32.0 (16)	5.2 (95% CI, 3.91–6.70)	5.8 (95% CI, 3.91–7.36)	4.1 (95% CI, 1.74–5.29)	0.49 (95% CI, 0.26–0.90; *p* = 0.034)
**Basal-like**	1.9 (5)	1.9 (4)	2.0 (1)	3.2 (95% CI, 1.51 to no result)	NA	NA	NA
**Normal-like**	14.2 (37)	16.3 (34)	6.0 (3)	6.8 (95% CI, 4.07–11.07)	NA	NA	NA
**All Luminals**	62.4 (163)	63.1 (132)	60.0 (30)	6.7 (95% CI, 4.40–8.05)	8.7 (95% CI, 6.67–11.07)	4.1 (95% CI, 2.66–4.37)	0.37 (95% CI, 0.26–0.52; *p* < 0.0001)
**All Non-luminals**	37.6 (98)	36.9 (77)	40.0 (20)	5.2 (95% CI, 3.45–6.70)	5.8 (95% CI, 3.91–8.18)	4.1 (95% CI, 2.76–5.29)	0.47 (95% CI, 0.26–0.85; *p* = 0.027)
**D**							
	**BOLERO-2 [15]** **All pts (n = 261)**						
	**% (n°)** **Overall Population**		**mPFS (Months)** **Overall Population**	**mPFS (Months)** **FULV + PALB**		**mPFS (Months)** **FULV + PBO**	**HR**
**Lum A**	44.0 (133)		NA	16.6		4.8	0.41 (95% CI, 0.25–0.66)
**Lum B**	30.8 (93)		NA	9.2		3.5	0.64 (95% CI, 0.38–1.09)
**HER2E**	20.9 (63)		NA	NA		NA	NA
**Basal-like**	1.7 (5)		NA	NA		NA	NA
**Normal-like**	2.6 (8)		NA	NA		NA	NA
**All Non-luminals**	25.2 (76)		NA	9.5		5.5	0.58 (95% CI, 0.34–0.99)
**E**							
	**MONALEESA-2 [24]**			**MONALEESA-3 [26]**			**MONALEESA-7 [28]**
	**All pts (n = 358)**			**All pts (n = 489)**			**All pts (n = 313)**
	**% (n°)**			**% (n°)**			**% (n°)**
**Lum A**	50.0 (179)			48.7 (238)			39.9 (125)
**Lum B**	28.8 (103)			19.6 (96)			25.2 (79)
**HER2E**	7.5 (27)			14.9 (73)			15.0 (47)
**Basal-like**	2.5 (9)			1.4 (7)			4.5 (14)
**Normal-like**	11.2 (40)			15.3 (75)			15.3 (48)
**F**							
	**All MONALEESA Trials [23]** **All pts (n = 1160)**						
	**% (n°)** **Overall Population**	**% (n°)** **Primary tumors** **(n = 835)**	**% (n°)** **Metastasis** **(N = 325)**	**mPFS (Months) Overall Population**	**mPFS (Months)** **ET + RIBO**	**mPFS (Months)** **ET + PBO**	**HR**
**Lum A**	46.7 (542)	49.9 (417)	38.1 (124)	NA	29.60 (95% CI, 23.03 to no result)	19.48 (95% CI, 15.61–24.80)	0.63 (95% CI, 0.49–0.83; *p* < 0.001)
**Lum B**	24.0 (278)	21.7 (181)	30.5 (99)	NA	22.21 (95% CI, 18.79 to no result)	12.85 (95% CI, 10.84–14.82)	0.52 (95% CI, 0.38–0.72; *p* < 0.001)
**HER2E**	12.7 (147)	11.2 (93)	16.8 (55)	NA	16.39 (95% CI, 12.71–24.60)	5.52 (95% CI, 3.12–9.17)	0.39 (95% CI, 0.25–0.60; *p* < 0.001)
**Basal-like**	2.6 (30)	2.4 (20)	2.9 (9)	NA	3.71 (95% CI, 1.91–13.00)	3.58 (95% CI, 1.87 to no result)	1.15 (95% CI, 0.46–2.83; *p* = 0.77)
**Normal-like**	14.0 (163)	14.8 (124)	11.7 (38)	NA	22.34 (95% CI, 16.56 to no result)	11.10 (95% CI, 7.39–16.56)	0.47 (95% CI, 0.30–0.72; *p* < 0.001)

# Biopsy source was unknown for two patients.

## Data Availability

Not applicable.

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
