# Peer review of "Role of Intrinsic Subtype Analysis with PAM50 in Hormone Receptors Positive HER2 Negative Metastatic Breast Cancer: A Systematic Review"

_ijms, 2022, doi:10.3390/ijms23137079_

Round 1

Reviewer 1 Report

Review of:

 Role of intrinsic subtype analysis with PAM50 in hormone receptors positive HER2 negative metastatic breast cancer: a systematic review

Comments to the Authors

General remarks:

The authors review five different publication with clinical trials, in which patients with HE+/HER2- metastatic breast cancers were enrolled. The authors aimed to correlate intrinsic subtype, treatment and outcome.

The five clinical trials patients are treated with inhibitors against aromatases, HER2, mTOR, CDK4/6, and Fulvestrant in various combinations. The authors describe each trial/publication extensively inside the text. It would have added much more clarity to present the various treatments and outcomes in a table, e.g. expanding table 1. I would recommend that.

Also, the five subtypes should be presented more in a table format. I am missing the Claudin-low subtype in this review, although it is an important one as described by various publication among others by the Perou group. This group should be added.

Generally, numbers higher than 12 are written in numerals and not as text (lines 137, 139, 163, 188, 232 …).

I would recommend an extensive editing of English language and style. Some parts are really hard to understand.

Specific comments:

Figure 1, I would have predicted a human search would have identified the same five publications and maybe even more by just searching clinical trials depositories by the various authorities.

Table 1, Expand this table or add another one with more information from the five publications.

Table 1A, Placebo treatments need to be added as a comparison otherwise there is no information about the benefit of the treatments.

Line 343 which Prat et al. publication does this refer to?

Conclusion, the authors state that the determination of intrinsic subtype cannot guide the choice of treatment (line332). I don’t think this is a valid statement as in the authors presented, treatments succeeding in various subtype increasing mPFS or OS. The authors state that as well in the next line 334, I would recommend rephrasing that first statement.

The authors mention the HARMONIA and the SOLTI-1718 NEREA trials, those should be discussed in the discussion rather than in the conclusion section.

I think it is very intriguing discussing the similarity between basal-like and triple negative BC. This could have been expanded upon and maybe a triple negative BC clinical trial is worth reviewing in this context.

Author Response

I take this opportunity to thank the reviewers for their valuable suggestions. Below I will list the main measures taken:

  • We expanded the tables by inserting all available data regarding percentage for each intrinsic subtype, outcome in the experimental and control arm. Unfortunately, these data were not equally available for all 5 studies considered.
  • As recommended, the sections regarding ongoing trials (HARMONIA and SOLTI-1718 NEREA) have been moved from conclusion to discussion.
  • The statement "determination of intrinsic subtype cannot guide the choice of treatment" has been reworded. The conclusions have been revised, also in consideration of the shifts towards the discussion of the previously mentioned paragraphs.
  • We agree that the similarity between basal-like and TNBC is very interesting. The concept was emphasized more in the discussion and conclusions. We also highlighted that the HARMONIA trial design includes an exploratory cohort of HR + / HER2- basal-like patients who will be treated with chemotherapy.
  • A reference to the claudin-low subgroup was inserted in the discussion, in the articles reported it had not been specifically analyzed.
  • Other minor corrections have been made and a linguistic review is in progress.

Reviewer 2 Report

The manuscript "Role of intrinsic subtype analysis with PAM50 in hormone receptors positive HER2 negative metastatic breast cancer: a systematic review" is a clinically interesting study based on clinical trials results.

The subject of research on HR positive and Her2 negative breast cancers is very important due to the high morbidity and high mortality rate of women with this cancer. Oncologists may find it helpful to have access to the collected test results.                                                                                     Due to the subject of the manuscript "Role of intrinsic subtype analysis with PAM50 in hormone receptors positive HER2 negative metastatic breast cancer: a systematic review", I propose to accept it for publication in the International Journal of Molecular Sciences in its current form.

Author Response

Thank you very much. We are very happy that you enjoyed our work.

Reviewer 3 Report

The authors presented conducted a systematic review to evaluate the role of the intrinsic subtype on the choice of treatment. The concluding statement in the abstract is that “at the moment, the intrinsic subtype has no decisive role in the choice of treatment, but it could acquire a prognostic and predictive role in the future.” This statement is not based on the systematic review, indicating that the currently no evidence but in the future it may. Overall, the review is poorly organized.

Major

·         Concluding remark: the concluding remark is not based on the review.

·         lack of quantitative linkage: the analysis is based on qualitative statistical consideration and no quantitative statistics is applied.

·         Results vs discussion: the current discussion is largely the summary of the discussion.

Minor

·         table 1: Table 1A-1C can be presented to highlight their similarity and difference.

Author Response

I take this opportunity to thank the reviewers for their valuable suggestions. Below I will list the main measures taken:

  • We expanded the tables by inserting all available data regarding percentage for each intrinsic subtype, outcome in the experimental and control arm. Unfortunately, these data were not equally available for all 5 studies considered.
  • The sections regarding ongoing trials (HARMONIA and SOLTI-1718 NEREA) have been moved from conclusion to discussion.
  • The conclusions have been revised, also in consideration of the shifts towards the discussion of the previously mentioned paragraphs.

Round 2

Reviewer 1 Report

The Authors’ changes greatly improved the manuscript. I still recommend an extensive editing of English language and style, writing numbers in numerals.

Author Response

The extensive editing of the English language and style is in progress, it will take a few more days, the revised file will be uploaded as soon as possible.
I take this opportunity again to thank you
for the suggestions that have contributed to improving this paper.

Reviewer 3 Report

As pointed out in the original review, the article presents no meaningful conclusion.  In the revised article, the concluding remark was removed based on the comment to the original manuscript since the concluding remark is not based on the review. Now, the revised manuscript does not provide any significant remark on the role of intrinsic subtype analysis with PAM50.

Author Response

Thanks for the tips. Our work collected and analyzed data in the literature. Some interesting points emerged in the discussion, which will need to be further explored, or in some cases they are already under evaluation (i.e. ongoing trials cited). However, this does not change our conclusion, currently intrinsic subtype determination is not used to guide therapeutic choice in clinical practice.